# Timely coupling of sleep spindles and slow waves linked to early amyloid-β burden and predicts memory decline

**Daphne Chylinski[1], Maxime Van Egroo[1], Justinas Narbutas[1,2], Vincenzo Muto[1], Mohamed Ali Bahri[1], Christian Berthomier[3], Eric Salmon[1,2,4], Christine Bastin[1,2], Christophe Phillips[1,5], Fabienne Collette[1,2], Pierre Maquet[1,4], Julie Carrier[6], Jean-Marc Lina[6], Gilles Vandewalle[1]***

[1]GIGA-Cyclotron Research Centre-In Vivo Imaging, University of Liège, Liège, Belgium; [2]Psychology and Cognitive Neuroscience Research Unit, University of Liège, Liège, Belgium; [3]Physip SA, Paris, France; [4]Department of Neurology, University Hospital of Liège, Liège, Belgium; [5]GIGA-In Silico Medicine, University of Liège, Liège, Belgium; [6]Centre for Advanced Research in Sleep Medicine, Université de Montréal, Montreal, Canada

**Abstract** Sleep alteration is a hallmark of ageing and emerges as a risk factor for Alzheimer's disease (AD). While the fine-tuned coalescence of sleep microstructure elements may influence age-related cognitive trajectories, its association with AD processes is not fully established. Here, we investigated whether the coupling of spindles and slow waves (SW) is associated with early amyloid-β (Aβ) brain burden, a hallmark of AD neuropathology, and cognitive change over 2 years in 100 healthy individuals in late-midlife (50–70 years; 68 women). We found that, in contrast to other sleep metrics, earlier occurrence of spindles on slow-depolarisation SW is associated with higher medial prefrontal cortex Aβ burden ($p=0.014$, $r^2_{\beta*}=0.06$) and is predictive of greater longitudinal memory decline in a large subsample ($p=0.032$, $r^2_{\beta*}=0.07$, N=66). These findings unravel early links between sleep, AD-related processes, and cognition and suggest that altered coupling of sleep microstructure elements, key to its mnesic function, contributes to poorer brain and cognitive trajectories in ageing.

**\*For correspondence:**
gilles.vandewalle@uliege.be

## Editor's evaluation

This paper is of interest to neuroscientists studying sleep, memory, and neurodegeneration. The authors found that an altered pattern of brain wave during NREM sleep, changes in the coupling of spindles and slow waves, correlates with amyloid burden and predicts memory decline over time in healthy older individuals. The results suggest that sleep brain waves may be a useful tool in identifying older adults at risk for future cognitive impairment in the earliest stage.

## Introduction

Alterations in sleep quality are typical of the ageing process with a more fragmented and less intense (or shallower) sleep detected as early as the fifth decade of life (*Carrier et al., 2011*). Beyond healthy ageing, alterations in sleep are predictive of the risk of developing Alzheimer's disease (AD) over the next 5–10 years (*Lim et al., 2013*; *Musiek and Holtzman, 2016*). Similarly, sleep disorders such as insomnia and obstructive sleep apnoea syndrome are associated with increased odds for AD diagnosis (*Baril et al., 2018*; *Elias et al., 2018*). Brain burdens of aggregated amyloid-β (Aβ) and tau proteins,

hallmarks of AD pathophysiology, have been linked with a reduced sleep intensity, as indexed by the overall production of slow waves (SW) during sleep, but also to worse objective sleep efficiency and subjective sleep quality, in healthy and cognitively normal older individuals aged >70 years (*Mander et al., 2015*; *Ju et al., 2017*; *Lucey et al., 2019*; *Kang et al., 2009*; *Ju et al., 2013*; *Sprecher et al., 2017*). Sleep alteration may, in turn, contribute to the aggregation of Aβ and tau proteins: experimental sleep deprivation and sleep fragmentation (disturbance in the production of SW during sleep) lead to increased concentration of Aβ in the CSF, both in animal models and in healthy human populations (*Ju et al., 2017*; *Kang et al., 2009*; *Lucey et al., 2018*; *Ooms et al., 2014*). Overall, a bidirectional detrimental relationship between sleep quality and the neuropathology of AD is emerging in the literature. Sleep may therefore constitute a modifiable risk factor which one could act upon to prevent or delay the neuropathological processes associated to AD and favour successful cognitive trajectories over the lifespan (*Van Egroo et al., 2019b*; *Ju et al., 2014*; *Wang and Holtzman, 2020*). Hitherto, however, sleep is not yet widely recognised as an independent risk factor for AD, and the mechanistic associations between sleep and early AD neuropathology are not yet fully established.

Sleep microstructure elements, such as sleep spindles and SW, are essential correlates of the cognitive functions of sleep, as higher density of both elements during post-learning sleep has been linked to a better overnight memory consolidation (*Ulrich, 2016*; *Gais et al., 2002*; *Schmidt et al., 2006*; *Mednick et al., 2013*; *Miyamoto et al., 2017*). Furthermore, the slower portion of SW activity (i.e. a power measure combining the density and amplitude of sleep SW over sleep cycles in the 0.6–1 Hz band) was reported to modulate the regression between the burden of Aβ over the prefrontal cortex and a lower overnight memory consolidation in cognitively normal older individuals (*Mander et al., 2015*). The fine-tuned coupling of spindles and SW has further been reported to be altered in ageing, with an earlier occurrence of the spindle relative to the SW depolarisation phase in the older compared to younger individuals, and to predict overnight memory retention (*Helfrich et al., 2018*). However, whether this change in the spindle-SW coupling in ageing is associated to AD-pathological processes and to individual cognitive trajectories is not fully established. A study in 31 individuals, aged around 75 years, found a link between the brain deposit of tau protein and the coupling of spindles and SW (*Winer, 2019*). By contrast, another research failed to find a link between this coupling and Aβ brain burden (*Winer et al., 2020*). Here, we argue that, on top of potential statistical power issues, the difficulty to detect this link may be due to the fact that the assessments were carried out late over the lifespan (i.e. >70 years), when subtle associations may be masked by concurrent brain alterations, and by the heterogeneity of sleep SW.

The low frequency oscillations of the electroencephalography (EEG) during non-Rapid Eye Movement (NREM) sleep have been divided in slow oscillations (≤1 Hz) and delta waves (1–4 Hz) for decades in humans, notably based on the theoretical framework of the generation of SWs (*Achermann and Borbély, 1997*; *Lee et al., 2004*; *Hubbard et al., 2020*). As one of the main features of the SW resides in their transition from down- to up-state, reflecting synchronised depolarisation, a recent work proposed the transition frequency of the down-to-up state as an objective way of distinguishing between slow and fast switcher SWs (*Bouchard et al., 2021*). Compared to young adults, on top of exhibiting a typical overall lower density of SW, older individuals reportedly exhibit higher probabilities of producing slow as compared to fast switcher SWs, providing important insights into age-related changes in sleep microstructure.

Investigating the coupling of spindles and SW, appropriately split between the slow and fast switcher SW, in late middle-aged healthy adults may be the best approach to gain insight into the biology underlying the early relationship between sleep and AD-related processes. In a longitudinal study, we therefore tested whether the coupling of spindles with the slow and the fast switcher SWs is differently associated with the early brain burden of Aβ in a large sample (N=100) of healthy and cognitively normal individuals in their late midlife (50–70 years). We further explored whether spindle coupling onto slow and fast switcher SWs would be associated with the change in memory performance at 2 years in a large subsample (N=66). We recorded habitual sleep in these individuals devoid of sleep disorders of both sexes (59.5±5 years; 68 women) under EEG and extracted the density and coupling of the spindles and fast and slow switcher SWs over frontal derivations. The burden of Aβ was assessed using PET tracers ([18F]Flutemetamol, N=97; [18F]Florbetapir, N=3) over the medial prefrontal areas, known to be an early site for Aβ deposits and the most important generator of SWs during sleep (*Mander et al., 2015*; *Dang-Vu et al., 2005*; *Dang-Vu et al., 2010*; *Saletin et al., 2013*).

Performance to the Mnemonic Similarity Task (MST), a memory task highly sensitive to early signs of cognitive decline (*Stark et al., 2013*; *Marks et al., 2017*) was assessed in all participants concomitantly to EEG and PET measurements as well as at follow-up, 2 years later (N=66). We hypothesised that our large sample of individuals, positioned relatively early in the ageing process, would allow to detect subtle differential associations between the impaired fine-tuned coupling of spindles and slow and fast switcher SWs, and both the early Aβ burden and memory performance decline over 2 years.

## Results

### Spindle onset on slow switcher SWs is linked to prefrontal Aβ burden

Following SW and spindle detections, we segregated SWs into those showing a slow or a fast down-to-up state transition, i.e., slow and fast switcher SWs (*Figure 1a*). Of the 341,836 detected slow switcher SWs, 75,910 co-occurred with a spindle (22%); while of the 78,235 fast switcher SWs, 26,912 (34%) were found to co-occur with a spindle. Regarding spindles, 563,928 spindles were detected over all the recordings, of which 102,822 were coupled to a SW (18%), 75,910 to slow switcher SWs (13%), and 26,912 to fast switcher SWs (5%). *Table 1* gathers average characteristics of SW types and spindles in addition the demographic characteristics of our sample.

Our primary analysis tested whether the phase of coupling of spindles onto SWs, as the dependent variable, was differently associated with the accumulation of Aβ protein over the medial prefrontal cortex (MPFC) (*Figure 1b*) for slow and fast switcher SWs, controlling for age, sex, and total sleep time (TST) as well as for SW type overall duration, as it significantly differed across SW types and could condition the time available for the co-occurrence of a spindle and an SW (*Table 1*). The generalised linear mixed model (GLMM) yielded a significant difference between SW types (main effect of SW type: $F_{1179.6}=34.97$, $p<0.0001$; $r^2_{\beta*}=0.16$) and no direct association with the burden of Aβ in the MPFC (main effect of Aβ burden: $F_{198.49}=0.09$, $p=0.76$), while, importantly, the interaction between the burden of Aβ in the MPFC and the type of SWs was significant (Aβ burden by SW type interaction: $F_{196}=7.05$, $p=0.009$; $r^2_{\beta*}=0.07$), controlling for all other covariates (age: $F_{197.3}=0.02$, $p=0.9$; sex: $F_{199.1}=.75$, $p=0.4$; TST: $F_{199.1}=0.75$, $p=0.4$, SW duration: $F_{1192.6}=8.73$, $p=0.0035$; $r^2_{\beta*}=0.04$). Post-hoc tests indicated that the link between the coupling of the spindles onto the SWs and the MPFC Aβ burden was significant for the slow switcher type ($t_{149.8}=-2.00$, $p=0.047$) and not for the fast switcher type ($t_{149.8}=0.56$, $p=0.57$) (*Figure 1c–d*). This primary result is in line with our initial hypothesis. In the following lines, we report additional statistical tests that were computed for a full understanding of this outcome.

We assessed whether spindles showed a preferential phase of anchoring with both slow and fast switcher SWs. Qualitative appreciation of the distributions suggests that there is no preferential phase of anchoring of spindles onto the fast switcher SWs, while spindle initiation onto slow switcher SWs would show a clear preferred phase (*Figure 1e*). Watson $U^2$ tests indicate, however, that the phase of anchoring onto slow and fast SWs is non-uniformly distributed (see Materials and methods; slow switcher SWs: $U^2=904.29$, $p<0.001$; fast switcher SWs: $U^2=136.76$, $p<0.001$), i.e., they both show some phase preference. Importantly, further statistical analysis with Watson's $U^2$ test showed that the distribution of spindles anchoring phase was significantly different between slow and fast switcher SWs ($U^2=71.143$, $p<0.001$). This finding reinforces the idea that slow and fast switcher SWs constitute distinct realisations of NREM oscillations that are differently associated with brain aggregation of Aβ during the ageing process. This has likely contributed to previous failures to detect links between the coupling of spindles and SWs and the deposit of Aβ. In fact, when testing the association between the coupling of spindles and SWs, irrespective of the type of SWs, and PET Aβ burden over the MPFC, the statistical analysis only yields a weak negative association between the phase of the coupling of the spindles onto the SWs and Aβ burden (main effect of Aβ uptake: $F_{196}=3.96$, $p=0.049$, $r^2_{\beta*}=0.04$; main effect of sex: $F_{196}=4.33$, $p=0.04$, $r^2_{\beta*}=0.04$; main effect of age: $F_{1,96}=0.05$, $p=0.83$), which could arguably go undetected in a smaller or different sample.

We further assessed the specificity of our main finding and found that the density of slow switcher SWs (main effect of Aβ PET uptake: $F_{196}=0.14$, $p=0.71$) or of spindles (main effect of Aβ PET uptake: $F_{196}=0.09$, $p=0.76$) was not associated with the MPFC Aβ burden (*Figure 1f–g*) after correcting for age ($F_{195}\leq3.71$, $p\geq0.06$), sex ($F_{195}\leq7.78$, $p\geq0.006$), and TST ($F_{195}\leq0.36$, $p\geq0.55$), further reinforcing the idea that it is the coupling of sleep microstructure elements that matters rather than their individual

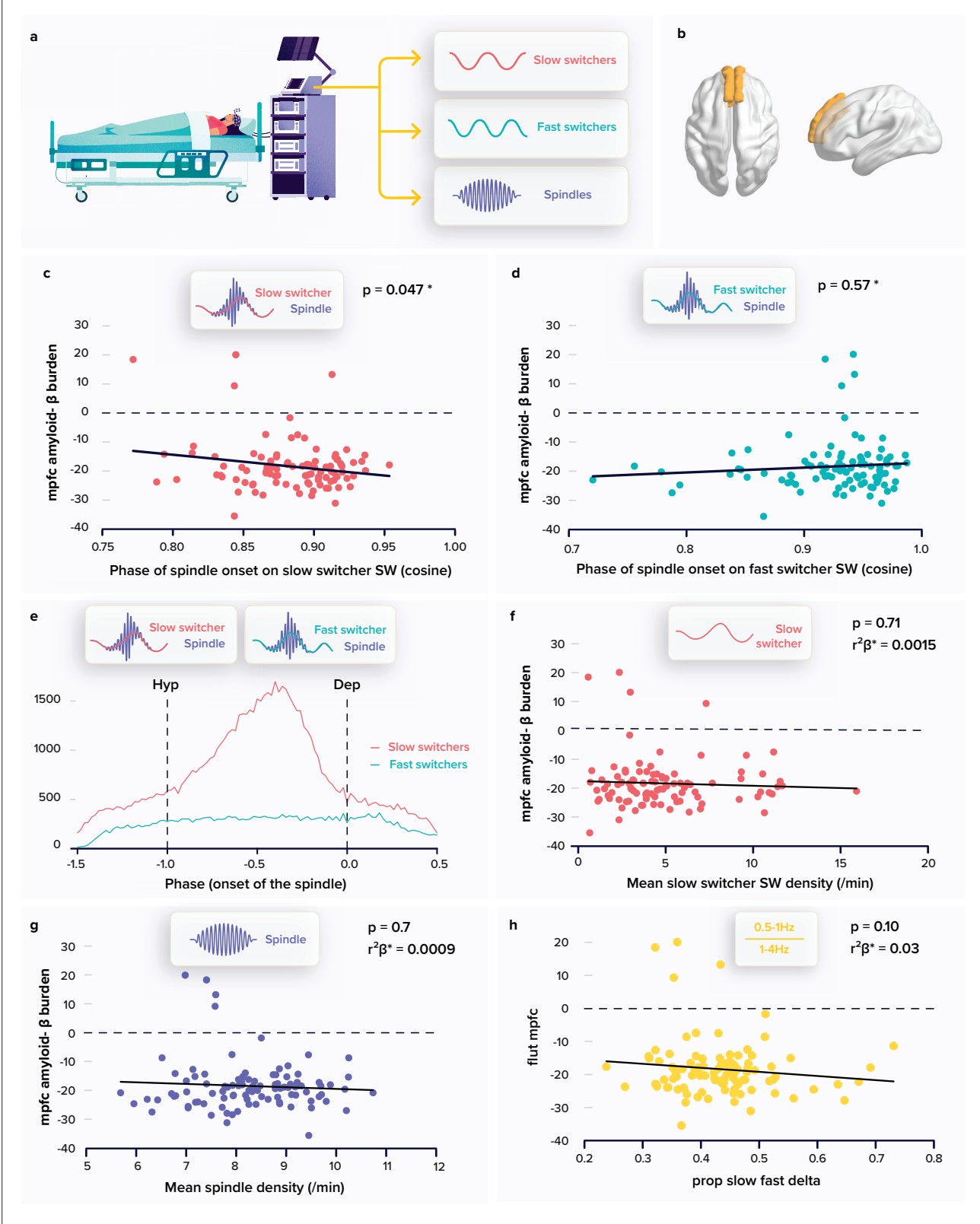

**Figure 1.** Relationships between spindle and slow wave (SW) metrics and the amyloid-β (Aβ) burden. (**a**) Following a screening night and a regular sleep-wake schedule for 1 week, the participants (N=100; 59.4±5.3 years; 68 women) slept in the lab at their habitual times under electroencephalography (EEG) recording. We extracted the density and coupling of spindles and fast and slow switcher SWs over frontal derivations during N2 and N3 sleep stage from EEG recordings. (**b**) PET signal uptake was measured over the medial prefrontal cortex (MPFC) depicted in yellow. (**c**) Significant negative

*Figure 1 continued on next page*

*Figure 1 continued*

association between the MPFC Aβ burden and spindle-slow switcher SW coupling. (**d**) No association between the MPFC Aβ burden and spindle-fast switcher SW coupling. (**e**) Analysis of the anchoring of the spindles onto the SWs yielded a difference in preferential coupling phase of slow (red) and fast switcher SWs (light blue) (the y axis represents the number of spindles starting at a specific SW phase). (**f**) No association between the MPFC Aβ burden and slow switcher SW density. (**g**) No association between the MPFC Aβ burden and spindle density. (**h**) No association between the MPFC Aβ burden and the ratio between .5-to-1 Hz over 1-to-4 Hz overnight cumulated EEG power. Except for *, p-values and $r^2\beta$* were computed from generalised linear mixed models (GLMMs) referred to in the text. Simple regressions were used only for a visual display and do not substitute the GLMM outputs. *Post-hoc test p-value is reported while Aβ burden-by-SW type interaction yielded p=0.009; $r^2\beta$*=0.07 (see text). We used the cosine value of the phase of coupling in the GLMMs (see Materials and Methods).

occurrence. Likewise, unlike previous reports (*Mander et al., 2015*; *Winer, 2019*), we did not find any association between the MPFC Aβ burden and the density of SWs, irrespective of their type ($F_{195}$=1.18, p=0.28) or slow wave energy (SWE) – the cumulated power generated in the delta band: ($F_{195}$=0.84, p=0.36), after correcting for age ($F_{195}$≥4.2, p≤0.04, $r^2_{\beta*}$≥0.04), sex ($F_{195}$≥6.63, p≤0.01, $r^2_{\beta*}$≥0.07), and TST ($F_{195}$≤0.36, p≥0.34). This is true also when attempting to reproduce a previous finding made in a smaller and older sample than in the present study (*Mander et al., 2015*): in our sample, power in the 0.5–1 Hz EEG frequency band as well as the ratio between the power in the 0.5–1 Hz and 1–4 Hz bands were not associated with the burden of Aβ (main effect of Aβ burden: $F_{195}$<2.85, p≥0.10) after correcting for age ($F_{195}$≥3.28 p=0.014, $r^2_{\beta*}$=0.03), sex ($F_{195}$≤0.25, p≥0.12), and TST ($F_{195}$≤2.7, p≥0.10) (*Figure 1h*).

**Table 1.** Sample characteristics.

| | Baseline (N=100) | Follow up (N=66) |
|---|---|---|
| Sex | 68 ♀ / 32 ♂ | 44 ♀ / 22 ♂ |
| Age (years) | 59.4±5.3 (50–69) | 59.9±5.4 (50–69) |
| Education (years) | 15.2±3.0 (9–25) | 14.9±3.3 (9–25) |
| Total sleep time (TST) (minutes, electroenchephalography [EEG]) | 392.8±45.9 (229–495.5) | 390.4±45.9 (264.0–495.5) |
| Time spent in N1 sleep stage (% of TST, EEG) | 6.2±2.8 (0.6–15.6) | 6.4±3.0 (0.6–15.6) |
| Time spent in N2 sleep stage (% of TST, EEG) | 51.6±8.9 (31.4–75.7) | 50.3±8.5 (32.8–75.7) |
| Time spent in N3 sleep stage (% of TST, EEG) | 19.2±6.4 (7.2–38.3) | 19.7±6.5 (8.2–38.3) |
| Time spent in Rapid Eye Movement (REM) sleep (% of TST, EEG) | 23.1±6.8 (6.5–39.8) | 23.6±7.4 (6.5–39.8) |
| Mean slow waves (SW) density (number/minute of N2/N3) | 7.1±4.3 (0.8–19.2) | 6.9±4.0 (1.0–19.2) |
| Slow switchers | 4.9±3.1 (0.6–15.9)*** | 4.9±3.1 (0.6–15.9)*** |
| Fast switchers | 2.1±1.6 (0.1–8.8) | 2.0±1.3 (0.3–5.7) |
| Mean SW amplitude (µV) | 101.5±12.4 (76.0–128.2) | 101.0±12.3 (76.0–125.2) |
| Slow switchers | 104.5±13.7 (77.7–131.3)*** | 104.0±13.5 (77.7–130.5)*** |
| Fast switchers | 94.0±11.0 (73.0–130.1) | 93.0±10.1 (73.5–113.7) |
| Mean SW duration (1/frequency) (ms) | 822±66 (619–966) | 821±63 (642–948) |
| Slow switchers | 906±46 (759–1,023)*** | 904±43 (807–1,003)*** |
| Fast switchers | 671±46 (552–794) | 670±45 (559–783) |
| Mean SW transition frequency | 1.4±0.1 (1.1–1.8) | 1.4±0.1 (1.1–1.8) |
| Slow switchers | 1.1±0.0 (1.0–1.2)*** | 1.1±0.0 (1.0–1.2)*** |
| Fast switchers | 2.0±0.1 (1.9–2.2) | 2.0±0.1 (1.9–2.2) |
| Mean spindle density (number/minute of N2/N3) | 8.3±1.1 (5.7–10.7) | 8.1±1.0 (6.0–10.2) |

BMI, alcohol consumption, anxiety, and depression scores as well as sleepiness levels of the sample can be found in ***Chylinski et al., 2021***. TST: total sleep time. Average values ± SD [range: min–max values]. ***A significant difference between slow and fast switcher SW at a p<0.001.

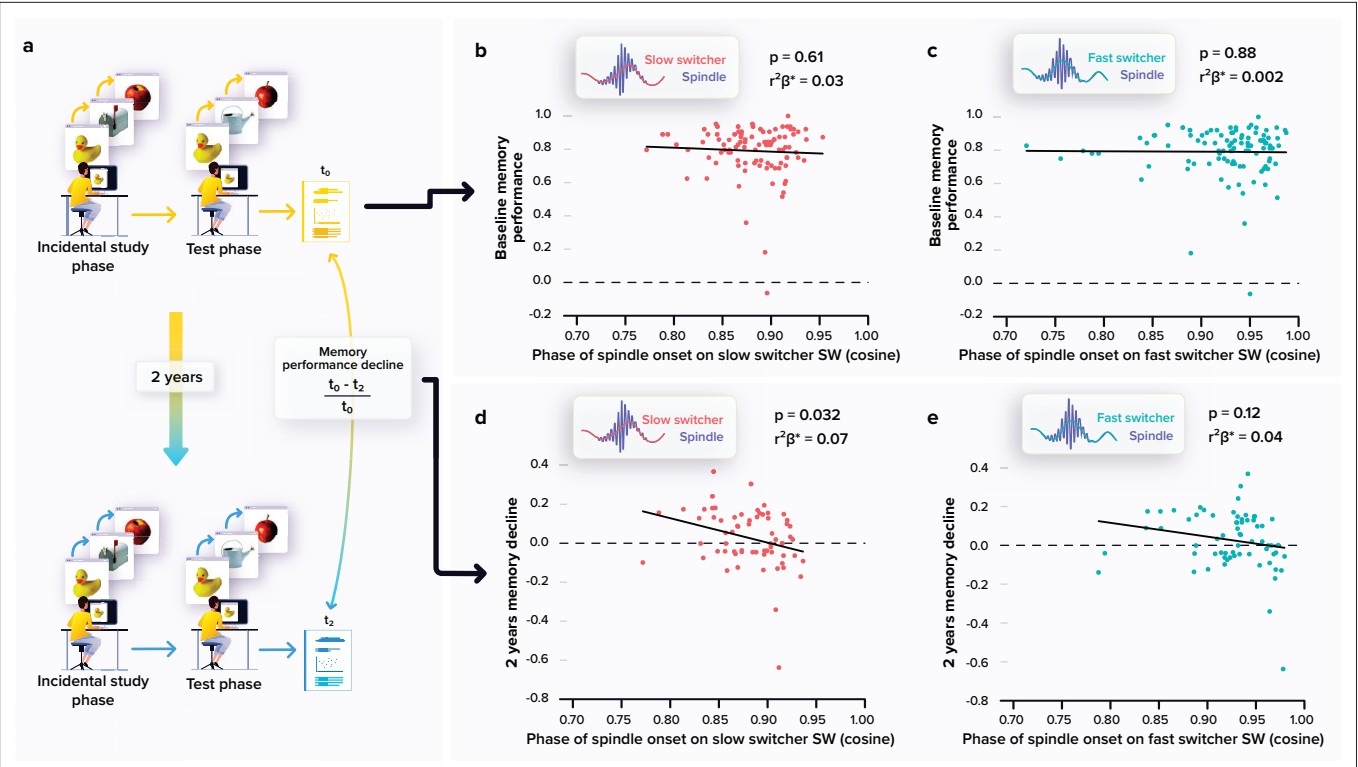

**Figure 2.** Relationships between memory performance and coupling between spindles and slow waves (SWs). (**a**) Memory performance was assessed through the Mnemonic Similarity Task (MST) where participants have to recognise previously encoded images in series of new or lure images (see Materials and methods). (**b**) No association between the baseline MST performance and spindle-slow switcher SW coupling. (**c**) No association between the baseline MST performance and spindle-fast switcher SW coupling. (**d**) Significant negative association between the 2 years relative change in MST performance and spindle-slow switcher SW coupling. (**e**) No association between the 2 years relative changes in MST performance and spindle-fast switcher SW coupling. p-values and $r^2\beta*$ were computed from generalised linear mixed models (GLMMs) referred to in the text. Simple regressions were used only for a visual display and do not substitute the GLMM outputs. We used the cosine value of the phase of coupling in the GLMMs (see Materials and methods).

### The anchoring of spindles onto slow-switcher SWs is associated to memory change over 2 years

A large subsample (N=66) completed a follow-up visit at 2 years which included the MST consisting in a pattern separation task targeting the ability to distinguish between highly resembling memory events, a hippocampus-dependent task which is very sensitive to early cognitive decline (*Stark et al., 2013*; *Marks et al., 2017*). We first observed an overall decline in performance between the baseline and follow-up performance at the MST ($t_{65}$=2.19, p=0.032). We then explored whether the coupling of spindles with the slow switcher SWs was associated with memory performance decline over 2 years (*Figure 2a*). Statistical analyses revealed a significant negative link between the relative change in memory performance and the phase of spindle anchoring onto slow switcher SWs, indicating that an earlier spindle onset is predictive of a memory worsening over 2 years (main effect of spindle-slow switcher SW coupling: $F_{161}$=4.80, p=0.032, $r^2_{\beta*}$=0.07), after correcting for age ($F_{161}$=0.25, p=0.62), sex ($F_{161}$=0.20, p=0.66), and education ($F_{161}$=0.25, p=0.62) (*Figure 2d–e*). No such association was detected when considering spindle coupling to fast switcher SWs (main effect of spindle-fast switcher SW coupling: $F_{161}$=2.51, p=0.12; main effect of age: $F_{161}$=0.33, p=0.57; main effect of sex: $F_{161}$=0.18, p=0.68; and main effect of education: $F_{161}$=1.11, p=0.30). Further statistical analyses show that the memory performance change over the 2 year was not significantly related to the MPFC Aβ burden (main effect of Aβ burden: $F_{160}$=2.33, p=0.13; main effect of age: $F_{160}$=1.27, p=0.26; main effect of sex: $F_{160}$=0.03, p=0.87; and main effect of education: $F_{160}$=0.41, p=0.53). When considering baseline performance to the MST across the entire sample (N=100; i.e. assessed at the same time as the sleep measures), we found no significant link between the coupling of the spindles onto both SW types

and the performance on the task (main effect of spindle-SW coupling: $F_{196} \geq 0.2$, $p \leq 0.61$; main effect of age: $F_{196} \leq 0.36$, $p \geq 0.55$; main effect of sex: $F_{196} \leq 0.54$, $p \geq 0.46$; and main effect of education: $F_{196} \leq 0.47$, $p \geq 0.50$ (*Figure 2b–c*)).

## Discussion

In order to unravel early associations between the microstructure of sleep and the burden of Aβ in the brain, and their cognitive implications, we collected polysomnography, PET, and behavioural data in a relatively large sample of individuals without cognitive impairments or sleep disorders. To this end, we recruited individuals in late middle age (50–70 years) that could in most cases only present limited age-related alterations in sleep and accumulation of Aβ protein in the brain (*Grothe et al., 2017*). We investigated whether the coupling of spindles onto SWs, showing a slower and a faster frequency of transition from the down to the up states (slow and fast switcher SWs), was associated to the accumulation of Aβ over the MPFC. We further explored whether the coupling of spindles onto SWs was associated with the performance to a sensitive memory test, assessed at the time of the sleep and PET recordings and, longitudinally, 2 years later. The coupling of spindles onto the slow, but not the fast, switcher SWs was significantly associated with the Aβ PET signal assessed over the MPFC. Moreover, this coupling between spindles and slow switcher SWs was significantly linked to the memory performance change detected 2 years after the initial assessment. Overall, our results provide compelling evidence that the link between sleep and the accumulation of Aβ over the MPFC, an early AD-related brain features, involves the precise and timely coupling of two key elements of NREM sleep, spindles and SWs, and that this coupling bears a predictive value for the subsequent decline in memory performance. Our study does not indicate, at least not in these healthy and relatively young older adults, that the number of spindles or SWs generated overnight is associated with the accumulation of Aβ over the MPFC.

Sleep SWs provide a readout of the homeostatic sleep pressure and are more prevalent at the beginning relative to the end of the night (*Achermann et al., 1993*). In addition, both the density of spindles and SWs have separately been related to overnight consolidation of memory (*Miyamoto et al., 2017*; *Fernandez and Lüthi, 2020*). They actively take part in information transfer from hippocampic to neocortical networks and in synaptic plasticity (*Ulrich, 2016*; *Miyamoto et al., 2017*). Recent research has put forward the importance of their precise phase coupling during sleep, and reported an age-related difference in that coupling (*Helfrich et al., 2018*). In the younger individuals, spindles tend to reach their maximum around the cortical up-state of the SWs, whereas in older individuals, spindles occur earlier on the depolarisation phase of the SWs. This earlier coupling between spindles and SWs is related to a poorer overnight memory retention (*Helfrich et al., 2018*), suggesting a suboptimal neuronal interplay for the exchange of information during sleep in older individuals. In line with a presumed suboptimal coupling between spindles and SWs, we find that, when controlling for age, individuals for which the spindles occur earlier during the transition phase of the slow switcher SWs show higher Aβ burden over the MPFC and a worse memory change over time. We did not find any significant effect of the age of participants on the phase of the coupling between spindles and SWs. This is likely due, in part, to the restricted age range of our participants, but also to the variability existing between the individuals in the changes they undergo in their sleep during ageing. Importantly, the relationship we observed between the phase of the spindle onset onto the SWs and the MPFC Aβ burden shows the same directionality (i.e. earlier spindles onto SWs) as the changes previously reported in older individuals.

Sleep SWs are classically divided into slow oscillations and delta waves based on whether their overall frequency lay between ~0.5 and 1 Hz or between 1 and 4 Hz, respectively (*Steriade, 2003*). Although meaningful, the definition of these frequency bands is arguably arbitrary. SWs were recently divided into two categories based on an objective functional feature consisting in the frequency of their transition from the down- to up-state, which reflects the relative synchronisation of the depolarisation of the neurons when generating a SW (*Bouchard et al., 2021*). Beyond the well-known decrease in the production of SWs in ageing, the slow switcher SWs were relatively preserved in the older individuals compared to the fast switcher SWs. This finding suggests that the two populations of SWs constitute distinct elements of sleep microstructure. Three present results confirm that the two types of SWs – slow and fast switchers – behave differentially. First, sleep spindles show a difference in their preferential coupling with the transition period from down-to-up state of the slow and fast

switcher SWs. While spindles occurring concomitantly to slow switchers SWs show a clear preference to the late part of the depolarisation phase, spindles co-occurring with fast switcher SWs show more widespread phase of coupling (that is still not randomly/uniformly distributed). Furthermore, only the coupling of spindles onto slow switcher SWs was significantly associated to the early accumulation of Aβ in the brain. Finally, only the coupling of spindles and slow switcher SWs was predictive of the memory change after 2 years. Our results support that slow switcher SWs, the type previously reported to be relatively spared during ageing (*Bouchard et al., 2021*), are important to the development of AD-related pathological changes, at least in the form of Aβ protein accumulation, and for the subsequent development of subtle alteration in cognitive abilities, leastways over the memory domain.

Sleep spindles are considered as thalamic events. They are generated through the interplay between the inhibitory reticular nucleus and the excitatory thalamocortical neurons, which project to the cortical neurons that feedback in turns to the thalamus (*Ulrich, 2016*). In contrast, SWs are intrinsically cortical. They consist in the spontaneous alternations between down (hyperpolarised) and up (depolarised) neuronal states (*Adamantidis et al., 2019*). The SWs undergo, however, an influence from the thalamus reticular nucleus that contributes to the synchronisation of distant neuronal populations (*Lee et al., 2004*; *Adamantidis et al., 2019*). Hence, although the exact neurophysiological origin of the functional associations between spindles and SWs remain to be established, the thalamus reticular neurons could arguably be involved. Our results would therefore indicate that the early aggregation of Aβ protein in the MPFC could disturb the thalamocortical interplay, driving the coalescence of spindles and SWs. This chronic disturbance would then trigger the cognitive changes detectable after 2 years. The cross-sectional nature of our imaging data does not, however, allow to make inferences regarding the directionality of the relationship between the spindle-slow switcher SW coupling and the burden of Aβ protein. Evidence accumulates to show that if AD-neuropathological hallmarks can affect the quality of sleep, sleep can also impinge onto these hallmarks (*Van Egroo et al., 2019b*). Specifically relevant in the context of this study, the occurrence of the SWs has been associated with a transient increase in the glymphatic flow related to local variations in neuromodulator concentrations (*Xie et al., 2013*). It is therefore possible that the changes in the density of SWs occurring during ageing affect both their coupling with spindles and the early accumulation of Aβ protein.

We found that the density of either spindles or SWs is not related to the early accumulation of the Aβ protein in the MPFC. This suggests a specific role for the coupling of both elements of sleep microstructure. Moreover and as previously reported in an intermediate analysis of a subsample of the same study (*Van Egroo et al., 2019a*), we do not find significant associations between more macroscopic measures, which are typically used to characterise sleep, and the accumulation of Aβ protein. The cumulated power of the oscillations generated over the entire 0.5–4 Hz delta band (i.e. the SWE) or only over the slower 0.5–1 Hz portion of SWE (as in *Mander et al., 2015*) as well as the ratio between slower (0.5–1 Hz) and faster (1–4 Hz) SWE (as in *Mander et al., 2015*) was not associated with PET measures. This finding argues against the idea that these rougher measures of sleep disruption constitute the earliest manifestations of the association between sleep and AD-related processes and contrasts with previous reports in a smaller sample of individuals older (>70 years) than our sample. Altogether, these discrepancies reinforce the idea that the alteration in the microstructure of sleep, consisting in the coupling of the spindles onto a specific subpopulation of SWs, as reported here, but also in the occurrence of microarousals during sleep we previously reported based on the same sample (*Chylinski et al., 2021*), shows a prior association with AD-related processes compared with the amount of slow brain oscillations generated during overnight sleep. The latter may only be significantly associated at a later age, when the pathophysiological changes are already more substantial. In addition, the coupling of spindles onto slow switcher SWs is predictive of the future change in memory performance. Sleep microstructure could therefore constitute a promising early marker of future-cognitive and brain-ageing trajectory (*Chylinski et al., 2021*). We did not evaluate whether distinct links between the SW types and slow and fast spindles are observed. As some reports describe that fast spindles are rather coupled to the up-state of the SWs and slow spindles tend to occur on the waning depolarisation phase of the SWs (*Mölle et al., 2011*), we could hypothesise that the associations we observe are probably rather driven by fast spindles. Future investigations are, however, needed to confirm this hypothesis.

One should bear in mind the potential limitations of our study. First, although we collected data in a relatively large sample, we may have insufficient power to detect some associations with other sleep measures. One can nevertheless frame our findings in relative terms such that association between spindle-SWs coupling and the early accumulation of Aβ protein is at least stronger than the association with the coupling of spindles onto fast switchers SW, the density of SWs and spindles, the SWE, etc. Also, effect sizes of the significant associations we found were relatively small, in line with the complex and multifactorial nature of AD. In addition, the longitudinal aspect of our study is relatively short-termed and only concerned the performance to a sensitive mnesic task while it did not include sleep EEG and the PET assessments. Further studies should replicate our findings and evaluate the predictive value of such parameters on longer longitudinal protocols, and the evolution of the sleep EEG and the PET parameters as well as their generalisability over other precociously impacted cognitive abilities. Finally, given that our protocol does not include manipulation of the coupling of the spindles onto the SWs, it precludes any inference on the causality of one aspect onto the other. It may be that cognition and the coupling of spindles and SWs are sensitive to the same age-related or AD-related phenomenon (e.g. presence of amyloid oligomers, or non-amyloid processes, that would go mostly undetected using common PET scan Aβ radioligand; *Yamin and Teplow, 2017*).

Together, our findings reveal that the timely occurrence of spindles onto a specific type of SWs showing a relative preservation in ageing may play an important role in ageing trajectory, both at the cognitive level and with regards to structural brain integrity. These findings may help to unravel early links between sleep, AD-related pathophysiology, and cognitive trajectories in ageing and warrants future clinical trials attempting at manipulating sleep microstructure or Aβ protein accumulation.

## Materials and methods
### Study design and participants

101 healthy participants aged from 50 to 70 years (68 women; mean ± SD = 59.4±5.3 years) were enrolled between 15 June 2016 and 2 October 2019 for a multimodal cross-sectional study taking place at the GIGA-Cyclotron Research Centre/In Vivo Imaging of the University of Liège (Cognitive fitness in ageing – COFITAGE – study) which has already led to several scientific publications (e.g. *Chylinski et al., 2021*; *Rizzolo et al., 2021*). One participant was excluded from analyses due to the lack of PET imaging data. The exclusion criteria were as follows: clinical symptoms of cognitive impairment (Mattis Dementia Rating Scale >130; Mini-Mental State Evaluation >27); recent psychiatric history, or severe brain trauma; self-reported or clinically diagnosed sleep disorder; ≤18 and ≥29; use of medication affecting sleep or the CNS; smoking; excessive alcohol (>14 units/week) or caffeine (>5 cups/day) consumption; shift work in the 6 months or transmeridian travel in the 2 months preceding the study. All participants gave their written informed consent prior to their participation. The study was registered with EudraCT 2016-001436-35. All procedures were approved by the Hospital-Faculty Ethics Committee of ULiège. All participants signed an informed consent prior to participating in the study.

### Sleep assessment

A first night of sleep was recorded at the laboratory under full polysomnography to avoid potential first night effects and exclude volunteers with sleep apnoea (Apnea Hypopnea Index ≥15 /hr). A second night of sleep was recorded with EEG, following 1 week of regular sleep-wake schedule based on each participant's preferred bed and wake-up time (compliance was verified by actimetry and sleep diary – Actiwatch, Cambridge Neurotechnology, UK). Sleep was recorded with N7000 amplifiers (EMBLA, Natus, Planegg, Germany). The recording comprised 11 EEG derivations, placed according to the 10–20 system (F3, Fz, and F4; C3, Cz, and C4; P3, Pz, and P4; O1 and O2), two bipolar electrooculogram, and two bipolar submental electromyogram electrodes. Sampling was set at 200 Hz, and the signal was re-referenced to the mean of the two mastoids. Recordings were scored for sleep stages in 30 s windows using a validated automatic algorithm (ASEEGA, Physip, Paris, France) (*Berthomier et al., 2007*; *Peter-Derex et al., 2021*). Automatic arousal and artefact detection (*Chylinski et al., 2020*; *'t Wallant et al., 2016*) were performed in order to remove EEG segments containing artefacts and arousals from further analysis.

## Slow wave and spindle detections

Only the frontal electrodes were considered because the frontal cortex is an early site showing Aβ deposit and is the primary generator of the SWs during sleep (*Mander et al., 2015*; *Dang-Vu et al., 2005*; *Dang-Vu et al., 2010*; *Saletin et al., 2013*) as well as to facilitate interpretations of future large-scale studies using headband EEG restricted to frontal electrodes (*Lucey et al., 2019*). SWs were automatically detected during N2 and N3 30s-epochs of NREM sleep devoid of artefacts/arousals >5 s long, using a previously developed algorithm (*Rosinvil et al., 2021*). Data were first band-filtered between 0.3 and 4.0 Hz with a linear phase finite impulse response (FIR) filter. Following recent work, SW detection criteria was adapted for age and sex (*Rosinvil et al., 2021*): peak to peak amplitude ≥70 μV (resp. ≥60.5 μV) and negative amplitude ≤–37 μV (resp. ≤–32 μV) was used for women (resp. for men), instead of the standard ≥75 μV and ≤–40 μV. The duration of the negative deflection had to fit in the range 125–1500 ms, and the duration of the positive deflection could not exceed 1000 ms. The SWs were sorted according to their transition frequency (*Bouchard et al., 2021*) (inverse of the duration between the hyperpolarised and depolarised state) into either slow or fast switchers (the critical value for distinguishing between the two types being the intersection between two Gaussian, around 1.2 Hz) (*Figure 3*).

Sleep spindles were also automatically detected over the same N2 and N3 epochs with a previously published methods (*Gaudreault et al., 2018*; *Lafortune et al., 2014*; *Martin et al., 2013*). The EEG signal was bandpass filtered between 10 and 16 Hz with a linear phase FIR filter (–3 dB at 10 and 16 Hz). The envelope amplitude of the Hilbert transform of this band-limited signal was smoothed, and a threshold was set at the 75th percentile. All events of duration between 0.5 and 3 s were then selected as a spindle.

After detection of SWs and spindles, analysis of their coincidence was performed. A coincidence was defined as to occurrence of the ignition of a spindle within the time frame of a SW: SW ignition at zero μV=phase 0°, SW maximum hyperpolarisation=π/2, zero crossing=π, SW maximum depolarisation=1.5π, and SW termination at zero μV=2π. This criterion was used on slow and fast switchers. All metrics related to SWs and spindles and to their coupling were averaged over F3, F4, and Fz derivations.

## MRI data

Quantitative multi-parametric MRI acquisition was performed on a 3-Tesla MR scanner (Siemens MAGNETOM Prisma, Siemens Healthineers, Erlangen, Germany). Quantitative maps were obtained by combining the images using different parameters sensitive to distinct tissue properties. The multi-parameter mapping was based on multi-echo 3D fast low angle shot at 1 mm isotropic resolution (*Weiskopf and Helms, 2008*). This included three datasets with T1, proton density (PD), and magnetisation transfer (MT)-weighted contrasts imposed by the choice of the flip angle (FA = 6° for PD and MT, 21° for T1) and the application of an additional off-resonance Gaussian-shaped Radio Frequency (RF) pulse for the MT-weighted acquisition. MRI multi-parameter maps were processed with the hMRI toolbox (*Tabelow et al., 2019*) (http://hmri.info) and SPM12 (Welcome Trust Centre for Neuroimaging, London, UK) to obtain notably a quantitative MT map, which was segmented into grey matter, white matter, and CSF using unified segmentation (*Ashburner and Friston, 2005*). Flow-field deformation parameters obtained from DARTEL spatial normalisation of the individual MT maps were applied to the averaged co-registered PET images (*Ashburner, 2007*). The volumes of interest were determined using the automated anatomical labelling atlas (*Tzourio-Mazoyer et al., 2002*).

## PET scan

Aβ PET imaging was performed using [18F] Flutemetamol, except for three volunteers for which [18F] Florbetapir was used. PET scans were performed on an ECAT EXACT +HR scanner (Siemens, Erlangen, Germany). Participants received a single dose of the radioligand in the antecubital vein (target dose 185±10% MBq); image acquisition started 85 min after the injection and consisted of four frames of 5 min, followed by a 10 min transmission scan using 68Ge line sources. Images were reconstructed using a filtered back-projection algorithm including corrections for the measured attenuation, dead time, random events, and scatter using standard software (Siemens ECAT – HR +V7.1, Siemens/CTI Knoxville, TN, USA). Individual PET average images were produced using all frames and were then manually reoriented according to MT-weighted structural MRI volumes and co-registered to

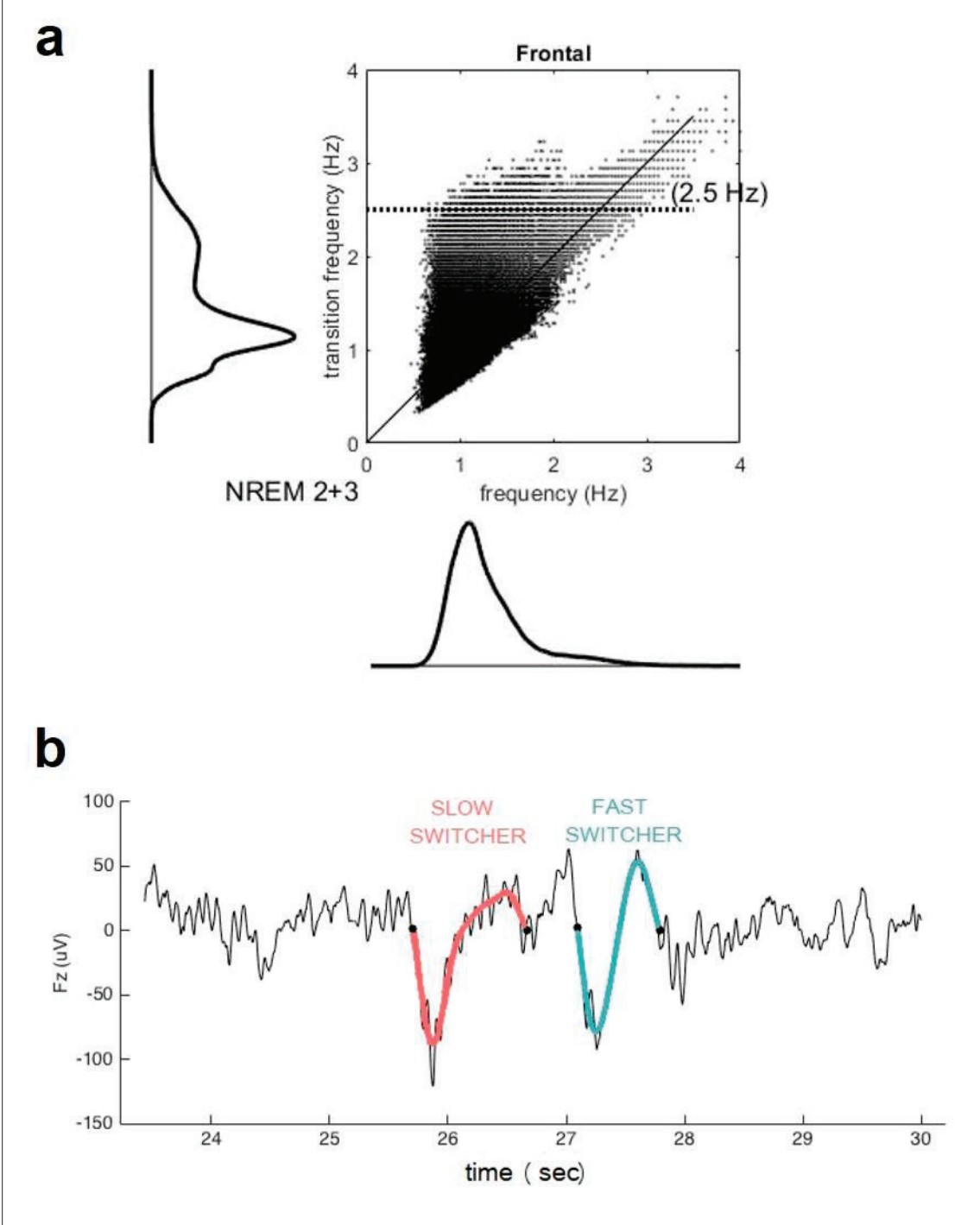

**Figure 3.** Transition frequency and mean frequency of slow waves (SWs). (**a**) Distribution of the mean frequency of the SWs (x axis) versus their transition frequency (y axis) for both NREM2 and NREM3 sleep stages in the entire study sample. One can observe a double distribution of the frequency of transition but not in the overall frequency. This shows that a faster or slower frequency of transition does not necessarily translate to an overall faster or slower SW. *Table 1* indicates, however, that duration of SW is significantly different between slow and fast switcher SWs. (**b**) Examples of slow switcher (red) and fast switcher SW (light blue) extracted from the electroencephalography signal for illustration purposes.

the individual space structural MT map. Standardised uptake value ratio (SUVR) was computed using the whole cerebellum as reference region (*Klunk et al., 2015*). As images were acquired using two different radioligands, their SUVR values were converted into Centiloid units (*Klunk et al., 2015*) (the validation of the procedure in our sample was previously published *Narbutas et al., 2021*). The Aβ

burden was averaged over a mask covering the MPFC, previously reported to undergo the earliest aggregation sites for Aβ pathology (*Grothe et al., 2017*).

## Cognitive assessments

As part of an extensive neuropsychological assessment, participants were administered the MST (*Stark et al., 2015*), a visual recognition memory task. After an incidental encoding phase during which participants were randomly presented 128 common objects for a period of 2 s and were instructed to determine whether the object presented on the screen was rather an 'indoor' or 'outdoor' item, the recognition memory phase consisted in the presentation of 192 objects (64 old, presented previously – target items; 64 similar but not identical to the previously presented stimuli – lure; and 64 new objects – foil items). In this phase, participants were instructed to determine whether the presented object was new (foil), previously presented (old), or similar but not perfectly identical (lure). For statistical analyses, the recognition memory (RM) score was used, computed as the difference between the rate of calling a target item 'old' minus the rate of calling a foil item 'old' (P['old'|target]-P['old'|foil]) (*Stark et al., 2013*; *Rizzolo et al., 2021*).

The MST was administered at two timepoints: the first time, the day preceding the baseline night, during a cognitive evaluation performed ~6.5 hr before habitual bedtime. The second neuropsychological evaluation was carried out ~24 months after the first one (mean 767±54 days) between 4 and 10h after wake up time. The memory decline score was computed as the baseline performance minus the follow-up performance, divided by the baseline performance, so that a higher score indicates a higher decline over the 2 years.

$$Memory\ decline = \frac{\text{RM baseline} - \text{RM follow-up}}{\text{RM baseline}}$$

## Statistics

Our primary analysis tested in a single model whether the phase of coupling, as dependent variable, would differ between SW types and would be associated with Aβ burden (independent variables), controlling for sex, age, TST, and SW duration. The phase of spindle-SW coupling was set as the phase of the onset of the spindle on the SW converted to its cosine value, to deal with the circularity of the phase variable and perform linear statistics (analysis using the phase in degrees yielded the same outcomes). As we tested our hypothesis in a single test, it did not require correction for multiple testing, and significance was set at $p < 0.05$. Our main exploratory analysis was to assess whether spindle-SW coupling to a specific SW type was associated with longitudinal change in the performance to the memory test, controlling for sex, age, education, and TST. The remaining analyses were aimed a better characterising, the primary analysis, or the main exploratory analyses.

Statistical analyses were performed using GLMMs in SAS 9.4 (SAS Institute, Cary, NC). The distribution of dependent variables was verified in MATLAB 2013a, and the GLMMs were adapted accordingly. Subject was treated as a random factor and each model was corrected for age and sex effects. Kenward-Roger's correction was used to determine the degrees of freedom. Cook's distance was used to assess the potential presence of outliers driving the associations, and as values ranged below 0.45, no data point was excluded from the analyses (a Cook's distant >1 is typically considered to reflect outlier value). Semi-partial $R^2$ ($R^2_{\beta*}$) values were computed to estimate the effect sizes of significant fixed effects and statistical trends in all GLMMs (*Jaeger et al., 2017*). p-values in post-hoc contrasts (difference of least square means) were adjusted for multiple testing using Tukey's procedure. Watson's non-parametric two-sample $U^2$ test for circular-normal data was performed in MATLAB 2019 to assess the difference between the distribution of spindle onset on the phase of SWs for slow and fast switcher SWs. We further assessed whether the distribution of spindle onset on the phase of SWs per type was different from a uniform distribution. For each SW type, we generated series of uniformly distributed random values composed of the same number of values spanning the same ranges. Watson's non-parametric two-sample $U^2$ test compared this random series to the actual values.

Optimal sensitivity and power analyses in GLMMs remain under investigation (e.g. *Kain et al., 2015*). We nevertheless computed a prior sensitivity analysis to get an indication of the minimum detectable effect size in our main analyses given our sample size. According to G*Power 3 (version 3.1.9.4) (*Faul et al., 2009*), taking into account a power of 0.8, an error rate α of 0.05, and a sample

size of 100 allowed us to detect small effect sizes $r > 0.27$ (two-sided; absolute values; CI: 0.07–0.44; $R^2 > 0.07$, $R^2$ CI:0.005–0.19) within a linear multiple regression framework including one tested predictor (Aβ) and two covariates (age and sex).

## Data availability

The data and analysis scripts supporting the results included in this manuscript are publicly available via the following open repository: https://gitlab.uliege.be/CyclotronResearchCentre/Public/fasst/slow-wave-spindle-coupling-and-amyloid; *Vandewalle, 2022*. We used Matlab script for MRI and PET data processing and to detect slow wave and spindles as well as their coupling, while we used SAS for statistical analyses. The raw data could be identified and linked to a single subject and represent a huge amount of data (>200 Gb). Researchers willing to access to the raw should send a request to the corresponding author (GV). Data sharing will require evaluation of the request by the local Research Ethics Board and the signature of a data transfer agreement (DTA).

## Acknowledgements

We thank M Blanpain, P Cardone, M Cerasuolo, E Lambot, P Ghaemmaghami, C Hagelstein, S Laloux, E Balteau, A Claes, C Degueldre, B Herbillon, P Hawotte, B Lauricella, A Lesoine, A Luxen, X Pepin, E Tezel, D Marzoli, C Schmidt, and P Villar González for their help in different steps of the study. This work was supported by Fonds National de la Recherche Scientifique (FRS-FNRS, FRSM 3451611, Belgium), Actions de Recherche Concertées (ARC SLEEPDEM 17/27–09) of the Fédération Wallonie-Bruxelles, University of Liège (ULiège), Fondation Simone et Pierre Clerdent, European Regional Development Fund (ERDF, Radiomed Project), and the Canadian Institutes of Health Research (CIHR) (grant number 190750). [18 F]Flutemetamol doses were provided and cost covered by GE Healthcare Ltd (Little Chalfont, UK) as part of an investigator sponsored study (ISS290) agreement. This agreement had no influence on the protocol and results of the study reported here MVE, CBastin, FC, CP, and GV are/were supported by the FRS-FNRS Belgium.

## Additional information

### Competing interests

Christian Berthomier: is an owner of Physip, the company that analysed the EEG data as part of a collaboration. This ownership and the collaboration had no impact on the design, data acquisition and interpretations of the findings. The other authors declare that no competing interests exist.

### Funding

| Funder | Grant reference number | Author |
| --- | --- | --- |
| Fonds De La Recherche Scientifique - FNRS | FRSM 3.4516.11 | Gilles Vandewalle |
| Fédération Wallonie-Bruxelles | ARC-SLEEPDEM 17/09 | Daphne Chylinski<br>Maxime Van Egroo<br>Justinas Narbutas<br>Christine Bastin<br>Christophe Phillips<br>Fabienne Collette<br>Pierre Maquet<br>Gilles Vandewalle |

| Funder | Grant reference number | Author |
|---|---|---|
| European Regional Development Fund | RAdiomed | Daphne Chylinski<br>Maxime Van Egroo<br>Justinas Narbutas<br>Vincenzo Muto<br>Mohamed Ali Bahri<br>Eric Salmon<br>Christine Bastin<br>Christophe Phillips<br>Fabienne Collette<br>Pierre Maquet<br>Gilles Vandewalle |
| Canadian Institutes of Health Research | grant number 190750 | Julie Carrier<br>Jean-Marc Lina |
| General Electric | ISS290 | Eric Salmon<br>Christine Bastin<br>Christophe Phillips<br>Fabienne Collette<br>Pierre Maquet<br>Gilles Vandewalle |
| Fonds De La Recherche Scientifique - FNRS | | Maxime Van Egroo<br>Christine Bastin<br>Christophe Phillips<br>Fabienne Collette<br>Gilles Vandewalle |
| Fondation Recherche Alzheimer | | Fabienne Collette |

The funders had no role in study design, data collection and interpretation, or the decision to submit the work for publication.

### Author contributions

Daphne Chylinski, Conceptualization, Data curation, Formal analysis, Investigation, Methodology, Writing - original draft, Writing – review and editing; Maxime Van Egroo, Justinas Narbutas, Data curation, Formal analysis, Investigation, Writing – review and editing; Vincenzo Muto, Formal analysis, Investigation, Methodology, Writing – review and editing; Mohamed Ali Bahri, Formal analysis, Methodology, Writing – review and editing; Christian Berthomier, Formal analysis, Resources; Eric Salmon, Conceptualization, Formal analysis, Funding acquisition, Resources, Writing – review and editing; Christine Bastin, Conceptualization, Formal analysis, Funding acquisition, Investigation, Writing – review and editing; Christophe Phillips, Conceptualization, Funding acquisition, Methodology, Resources, Writing – review and editing; Fabienne Collette, Conceptualization, Formal analysis, Funding acquisition, Investigation, Supervision, Writing – review and editing; Pierre Maquet, Conceptualization, Formal analysis, Funding acquisition, Writing – review and editing; Julie Carrier, Conceptualization, Formal analysis, Writing – review and editing; Jean-Marc Lina, Conceptualization, Formal analysis, Resources, Supervision, Writing - original draft, Writing – review and editing; Gilles Vandewalle, Conceptualization, Formal analysis, Funding acquisition, Investigation, Methodology, Project administration, Supervision, Writing - original draft, Writing – review and editing

### Author ORCIDs

Daphne Chylinski http://orcid.org/0000-0002-7319-0859
Eric Salmon http://orcid.org/0000-0003-2520-9241
Christine Bastin http://orcid.org/0000-0002-4556-9490
Christophe Phillips http://orcid.org/0000-0002-4990-425X
Julie Carrier http://orcid.org/0000-0001-5311-2370
Gilles Vandewalle http://orcid.org/0000-0003-2483-2752

### Ethics

Human subjects: The study was registered with EudraCT 2016-001436-35. All procedures were approved by the Hospital-Faculty Ethics Committee of ULiège. All participants signed an informed consent prior to participating in the study.

**Decision letter and Author response**
Decision letter https://doi.org/10.7554/eLife.78191.sa1
Author response https://doi.org/10.7554/eLife.78191.sa2

---

## Additional files

### Supplementary files
• Transparent reporting form

### Data availability
The data and analysis scripts supporting the results included in this manuscript are publicly available via the following open repository: https://gitlab.uliege.be/CyclotronResearchCentre/Public/fasst/slow-wave-spindle-coupling-and-amyloid (copy archived at swh:1:rev:a63699b5f06c98ade284689af5671e13c77751f2). We used Matlab script for MRI and PET data processing and to detect slow waves and spindles as well as their coupling, while we used SAS for statistical analyses. The raw data could be identified and linked to a single subject and represent a huge amount of data (> 200 Gb). Researchers willing to access the raw data should send a request to the corresponding author (GV). Data sharing will require evaluation of the request by the local Research Ethics Board of the Faculty of Medicine at the University of Liège, Belgium and the signature of a data transfer agreement (DTA).

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
