## [Editor Report]

This paper is of interest to neuroscientists studying sleep, memory, and neurodegeneration. The authors found that an altered pattern of brain wave during NREM sleep, changes in the coupling of spindles and slow waves, correlates with amyloid burden and predicts memory decline over time in healthy older individuals. The results suggest that sleep brain waves may be a useful tool in identifying older adults at risk for future cognitive impairment in the earliest stage.

---

## [Decision Letter]

**Decision letter after peer review:**

Thank you for submitting your article "Timely sleep coupling: spindle-slow wave synchrony is linked to early amyloid-β burden and predicts memory decline" for consideration by *eLife*. Your article has been reviewed by 3 peer reviewers, including Sakiko Honjoh as Reviewing Editor and Reviewer #1, and the evaluation has been overseen by Jeannie Chin as the Senior Editor.

Essential revisions:

1) One of the main conclusions of the manuscript is that spindles are preferentially coupled to slow switcher-SWs over fast switcher-SWs. Since spindle-SW coupling is defined as a co-occurrence of a spindle over a slow wave, the longer a slow wave is, the more co-occurrence would be expected. In figure 4b, fast switcher-SWs seem to show not only faster transition but also shorter duration. Therefore, the authors should include average slow wave durations of fast and slow switchers in table 1, and the analysis should control for the difference in SW durations in Figure 1.

2) The authors claim that the results of Mander et al., 2015 were never replicated in the cover letter. However, the introduction about this paper (ref 6) is misleading (lines 50-53), and the authors do not try to replicate their results. Mander et al., showed that 0.6-1 Hz and 1-4 Hz showed the opposite trends, negative and positive correlation with Aβ burden, respectively. Analyzing 0.3-4Hz altogether will cancel out their opposite effects. Therefore, to show the uncoupling of spindles and slow switcher SWs is an earlier and more sensitive marker of Aβ burden, the authors should analyze 0.6-1 Hz and 1-4 Hz separately in their own data.

3) The statistics part needs further details. Please clarify what are primary analyses and exploratory analyses, and adjust for multiple comparisons accordingly.

*Reviewer #1 (Recommendations for the authors):*

Though the data support the authors’ claim, it still remains unclear whether the uncoupling of spindles and slow switcher-SWs is the earliest marker since the authors did not analyze the 0.6-1 Hz frequency band used in reference 6. To show the uncoupling is an earlier marker than 0.6-1 Hz SWA, either correspondence of slow switchers to slow oscillation (0.6-1 Hz) or direct relationships among 0.6-1 Hz power, Aβ burden, and cognitive decline should be tested.

*Reviewer #2 (Recommendations for the authors):*

– How is ‘SW type’ entered into the model? Does this report the % of SWs of a single type for each participant?

– The methods for spindle detection are important for assessing and understanding the paper and it would be helpful to describe at least briefly in the Methods, even if it repeats prior work.

– Table 1 should define what is reported in +/- vs. brackets.

– I think the periods in the numbers for the first section of Results are intended to be commas, to indicate thousands?

– For the difference between fast and slow SWs, it would be helpful to either direct statistically test this, or else to make more cautious statements about the two SW types and acknowledge this limitation.

*Reviewer #3 (Recommendations for the authors):*

Suggestions:

– Clarify exactly what are the primary analyses of this study, explain why these are the primary outcomes, and adjust for multiple comparisons accordingly. The primary analyses and primary conclusions, should, ideally, be aligned.

– Explain in greater detail how the primary measure of spindle-slow wave coupling (phase angle) was decided upon, rather than, for instance, percent coincidence, or dispersion of phase angle as a measure of the “tightness” of coupling.

– Temper conclusions of causality given the observational nature of the work, and discuss in greater depth the likelihood that alterations in phase relationship may be markers of early AD-related brain changes, not picked up on by amyloid PET (e.g. amyloid oligomers, or non-amyloid processes).

– To what extent do relations with cognitive decline differ between those with high and low baseline amyloid burden?

---

## [Author Response]

Essential revisions:1) One of the main conclusions of the manuscript is that spindles are preferentially coupled to slow switcher-SWs over fast switcher-SWs. Since spindle-SW coupling is defined as a co-occurrence of a spindle over a slow wave, the longer a slow wave is, the more co-occurrence would be expected. In figure 4b, fast switcher-SWs seem to show not only faster transition but also shorter duration. Therefore, the authors should include average slow wave durations of fast and slow switchers in table 1, and the analysis should control for the difference in SW durations in Figure 1.

The transition frequency only concerns the part of the Down—Up transition of the slow wave. On Figure 3a (former figure 4a), one can see that although the distribution of transition frequency of SW reflects two distinct Gaussian distributions (vertical axis) the overall frequency of the SW was composed of single Gaussian (horizontal axis). This shows that a faster frequency of transition does not necessarily translate to an overall faster SW. There are slow slow waves with fast transition frequency and vice versa.

Yet, table 1 (PAGE 6) now includes average duration (1/frequency) for each SW type over our sample (overall duration of slow switcher SW is 906 ± 46 [759 – 1023] and of fast switcher is 671 ± 46 [552 – 794]) and a 2-sample t-test shows that duration is shorter for fast vs. slow switcher SWs (p <.001). Following this insightful suggestion, we included SW type duration in our main analyses and still get a significant SW type x Aβ interaction when using spindle-SW coupling as dependent variable [F_1,97.87_=6.42; p = 0.013], as now reported in the main text.

We modified the legend of the figure as follows to clarify this point (PAGE 17):

“Figure 3. Transition frequency and mean frequency of SWs.

a. Distribution of the mean frequency of the slow waves (x axis) versus their transition frequency (y axis) for both NREM2 and NREM3 sleep stages in the entire study sample. One can observe a double distribution of the frequency of transition but not in the overall frequency. This shows that a faster or slower frequency of transition does not necessarily translate to an overall faster or slower SW. Table 1 indicates, however, that duration of SW is significantly different between slow and fast switcher SWs.

*b.* Examples of slow switcher (red) and fast switcher SW (light blue) extracted from the EEG signal for illustration purposes.”

Please see Essential comment (3) for detailed changes made to the results and methods section.

2) The authors claim that the results of Mander et al., 2015 were never replicated in the cover letter. However, the introduction about this paper (ref 6) is misleading (lines 50-53), and the authors do not try to replicate their results. Mander et al., showed that 0.6-1 Hz and 1-4 Hz showed the opposite trends, negative and positive correlation with Aβ burden, respectively. Analyzing 0.3-4Hz altogether will cancel out their opposite effects. Therefore, to show the uncoupling of spindles and slow switcher SWs is an earlier and more sensitive marker of Aβ burden, the authors should analyze 0.6-1 Hz and 1-4 Hz separately in their own data.

The reviewer is partially correct and our original text was unclear. Mander et al., 2015 found a significant negative association between power in.6-1Hz frequency band (p=.032) and the ratio between of powers over.6-1Hz and 1-4 Hz (p=.02) and mPFC Aβ burden but the association was not significant when considering 1-4Hz alone (p=.076), precluding interpretation of the result. We had included the ratio between.6-1Hz and 1-4Hz in our initial submission and report that it was not associated with early Aβ burden (Page 7, line 134, of original text), but it was maybe “hidden” in the text.

We slightly modified the introduction to take this point into account (PAGE 3):

“Furthermore, the slower portion of SW activity (i.e. a power measure combining the density and amplitude of sleep SW over sleep cycles in the 0.6 to 1 Hz band) was reported to modulate the regression between prefrontal cortex Aβ burden and a lower overnight memory consolidation in cognitively normal older individuals”.

We further modified the result section to clarify this point (PAGE 9):

“Likewise, unlike previous reports, we did not find any association between the MPFC Aβ burden and several characteristics of the SWs (*SW density* – per min of NREM sleep: F_1,95_=1.18, p=0.28; *spindle density* – per min of NREM sleep: F_1,95_=0.06, p=0.80; *slow wave energy (SWE) –* cumulated power generated in the δ band – F_1,95_=0.84, p=0.36), after correcting for age, sex and total sleep time (SW density: age: F_1,95_=5.49, p=0.02, r²β*=0.05; sex: F_1,95_=10.26, p=0.002, r²β*= 0.1; TST: F_1,95_=0.5, p=0.48; spindle density: age: F_1,95_=0.56, p=0.46, sex: F_1,95_=0.42, p=0.52, TST: F_1,95_=0.36, p=0.55; SWE: age: F_1,95_=4.2, p=0.04, r²β*=0.04; sex: F_1,95_=6.63, p=0.01; TST: F_1,95_=0.9, p=0.34). This is true also when attempting to reproduce a previous finding made in a sample smaller and older than in the present study:6 in our sample, power in.5-1 Hz EEG frequency band as well as the ratio between the power in the.5-1Hz and 1-4 Hz bands were not associated with burden of Aβ (main effect of Aβ burden: F_1,95_ < 2.85, p ≥ 0.10) after correcting for age (F_1,95_ ≥ 3.28 p=0.014, r²β*=0.03), sex (F_1,95_ ≤ 0.25, p ≥ 0.12) and total sleep time (TST: F_1,95_ ≤ 2.7, p ≥ 0.10) (FIGURE 1i).”

We modified the discussion to clarify what Mander et al., 2015 had found as it was more appropriate than in the introduction (PAGE 13):

“The cumulated power of the oscillations generated over the entire.5-to-4 Hz δ band (i.e. the slow wave energy – SWE) or only over the slower.5-to-1 Hz portion of SWE (as in Mander et al., 2015) as well as the ratio between slower (.5-1Hz) and faster (1-4Hz) SWE (as in Mander et al., 2015) were not associated with PET measures.”

3) The statistics part needs further details. Please clarify what are primary analyses and exploratory analyses, and adjust for multiple comparisons accordingly.

Our primary analysis tested in a single GLMM whether the phase of coupling as dependent variable, would differ between SW types and would be associated with Aβ burden (independent variables), controlling for sex, age, TST and SW type duration (the latter covariate was added in response to a reviewer’s comment). As we tested our hypothesis in a single test it does require correction for multiple testing. In our initial submission, we felt that our results would be clearer by first reporting whether spindle coupling to SW varied with SW type and then reporting Aβ associations with each SW type prior to reporting our primary analyses. We understand that this may have been misleading.

Our main exploratory analysis was to assess whether spindle-SW coupling to a specific SW type was associated with longitudinal change in the performance to the memory test. Again, we felt that first reporting the cross-sectional association (not significant) and then the longitudinal change was clearer to the reader. We recognise that this may have been misleading.

We modified the end of the introduction as follows (PAGES 4-5):

“In a longitudinal study, we therefore tested whether the coupling of spindles with the slow and the fast switcher SWs is differently associated with the early brain burden of Aβ in a large sample (N=100) of healthy and cognitively normal individuals in late midlife (50-70y). We further explored whether spindle coupling onto slow and the fast switcher SWs would be associated the change in memory performance at 2 years in a large subsample. We recorded habitual sleep in these individuals devoid of sleep disorders of both sexes (59.5±5y; 68 women) under EEG and extracted the density and coupling of spindles and fast and slow switcher SWs over frontal derivations. The burden of Aβ was assessed using Positron Emission Tomography (PET) tracers ([18F]Flutemetamol, N=96; [18F]Florbetapir, N=4) over the medial prefrontal areas, known to be an early site for Aβ deposits and the most important generator of SWs during sleep. Performance to the Mnemonic Similarity Task (MST), a memory task highly sensitive to early signs of cognitive decline, was assessed in all participants concomitantly to EEG and PET measurements as well as at follow-up, 2 years later (N=66). We hypothesised that our large sample of individuals, positioned relatively early in the ageing process, would allow to detect subtle differential associations between the impaired fine-tuned coupling of spindles and slow and fast switcher SWs, and both the early Aβ burden and memory performance decline over 2 years.”

We modified the result section as follows (PAGES 5-10; please also see the new figure 1 and table 1):

“Spindle onset on slow switcher SW is linked to prefrontal Aβ burden

Following SW and spindle detection, we segregated SWs into those showing a slow or a fast down-to-up state transition, i.e. slow and fast switcher SWs (Figure 1a). Of the 341,.836 detected slow switcher SWs, 75,910 co-occurred with a spindle (22%); while of the 78,.235 fast switcher SWs, 26,912 (34%) were found to co-occur with a spindle. Regarding spindles, 563,928 spindles were detected over all the recordings, of which 102,822 were coupled to a SW (18%), 75,910 to slow switcher SWs (13%) and 26.912 to fast switcher SWs (5%). Table 1 gathers average characteristics of SW types and spindles in addition the demographic characteristics of our sample.

Our primary analysis tested whether the phase of coupling of spindles onto SWs, as the dependent variable, was differently associated with the accumulation of Aβ protein over the MPFC (Figure 1b) for slow and fast switcher SWs, controlling for age, sex, and TST as well as for SW type overall duration, as it significantly differed across SW types and could condition the time available for the co-occurrence of a spindle and a SW (Table 1). The GLMM yielded a significant difference between SW types (main effect of SW type: F_1,179.6_= 34.97, p <0.0001; r²_β*_=0.16) and no direct association with the burden of Aβ in the MPFC (main effect of Aβ burden: F_1,98.49_=0.09, p = 0.76), while, importantly, the interaction between the burden of Aβ in the MPFC and the type of SWs was significant (Aβ burden by SW type interaction: F_1,96_=7.05**,** p=0.009; r²_β*_=0.07), controlling for all other covariates (age: F_1,97.3_=.02, p=0.9**;** sex: F_1,99.1_=.75, p=0.4; TST: F_1,99.1_=.75, p=0.4, SWs duration: F_1,192.6_=8.73, p = 0.0035; r²_β*_= 0.04). *Post-hoc* tests indicated that the link between the coupling of the spindle onto the SW and the MPFC Aβ burden was significant for the slow switcher type (t_149.8_=-2.00, p=0.047) and not for the fast switcher type (t_149.8_=0.56, p=0.57) (Figure 1c-d). This primary result is in line with our initial hypothesis. In the following lines, we report additional statistical tests that were computed for a full understanding of this outcome.

We assessed whether spindles showed a preferential phase of anchoring with both slow and fast switcher SWs. Qualitative appreciation of the distributions suggests that there is no preferential phase of anchoring of spindles onto the fast switcher SWs while spindle initiation onto slow switcher SWs would show a clear preferred phase (Figure 1e). Watson U² tests indicate, however, that the phase of anchoring onto slow and fast SWs are both non-uniformly distributed (see methods; slow switcher SWs: U² = 904.29, p <0.001; fast switcher SWs: U² = 136.76, p <0.001), i.e. they both show some phase preference. Importantly, further statistical analysis with Watson’s U² test showed that the distribution of spindles anchoring phase was significantly different between slow and fast switcher SWs (U² = 71.143, p <0.001). This finding reinforces the idea that slow and fast switcher SWs constitute distinct realisations of NREM oscillations that are differently associated with brain aggregation of Aβ during the ageing process. This has likely contributed to previous failures to detect links between the coupling of spindles and SW and the deposit of Aβ. In fact, when testing the association between the coupling of spindles and SWs, irrespective of the type of SWs, and PET Aβ burden over the MPFC, the statistical analysis only yields a weak negative association between the phase of the coupling of the spindle onto the SW and Aβ burden (main effect of Aβ uptake: F_1,96_=3.96, p=0.049, r²_β*_=0.04; main effect of sex: F_1,96_=4.33, p=0.04, r²_β*_=0.04; main effect of age: F_1,96_=0.05, p=0.83), which could arguably go undetected in a smaller or different sample.

We further assessed the specificity of our main finding and found that the density of slow switcher SWs (main effect of Aβ PET uptake: F_1,96_=0.14, p=0.71) or of spindles (main effect of Aβ PET uptake: F_1,96_=0.09, p=0.76) was not associated with the MPFC Aβ burden (Figure 1f-g) after correcting for age (F_1,95_ ≤ 3.71, p ≥ 0.06), sex (F_1,95_ ≤ 7.78, p ≥ 0.006) and total sleep time (F_1,95_ ≤ 0.36, p ≥ 0.55), further reinforcing the idea that it is the coupling of sleep microstructure elements that matters rather than their individual occurrence. Likewise, unlike previous reports,^6,23^ we did not find any association between the MPFC Aβ burden and the density of SW, irrespective of their type (F_1,95_=1.18, p=0.28) or slow wave energy (SWE) – the cumulated power generated in the δ band: (F_1,95_=0.84, p=0.36), after correcting for age (F_1,95_ ≥ 4.2, p ≤ 0.04, r²_β*_ ≥ 0.04);, sex (F_1,95_ ≥ 6.63, p ≤ 0.01, r²_β*_ ≥ 0.07) and total sleep time (F_1,95_ ≤ 0.36, p ≥ 0.34). This is true also when attempting to reproduce a previous finding made in a sample smaller and older than in the present study:^6^ in our sample, power in.5-1 Hz EEG frequency band as well as the ratio between the power in the.5-1Hz and 1-4 Hz bands were not associated with the burden of Aβ (main effect of Aβ burden: F_1,95_ < 2.85, p ≥ 0.10) after correcting for age (F_1,95_ ≥ 3.28 p=0.014, r²_β*_=0.03), sex (F_1,95_ ≤ 0.25, p ≥ 0.12) and total sleep time (TST: F_1,95_ ≤ 2.7, p ≥ 0.10) (FIGURE 1h).

Slow switcher spindle phase coupling is associated to memory change over two years

A large subsample (N=66) completed a follow-up visit at 2 year which included the MST consisting in a pattern separation task targeting the ability to distinguish between highly resembling memory events, a hippocampus dependent task which is very sensitive to early cognitive decline.^32,33^ We first observed an overall decline in performance between the baseline and follow-up performance at the MST (t_65_ = 2.19, p=0.032). We then explored whether the coupling of spindles with the slow switcher SWs was associated with memory performance decline over 2 years (Figure 2a). Statistical analyses revealed a significant negative link between the relative change in memory performance and the phase of spindle anchoring onto slow switcher SWs, indicating that an earlier spindle onset is predictive of a memory worsening over two years (main effect of spindle-slow switcher SW coupling: F_1,61_=4.80, p=0.032, r²_β*_=0.07), after correcting for age (F_1,61_=0.25, p=0.62), sex (F_1,61_=0.20, p=0.66) and education (F_1,61_=0.25, p=0.62) (Figure 2d-e). No such association was detected when considering spindle coupling to fast switcher SWs (main effect of spindle-fast switcher SW coupling: F_1,61_=2.51, p=0.12; main effect of age: F_1,61_=0.33, p=0.57, main effect of sex: F_1,61_=0.18, p=0.68, main effect of education: F_1,61_=1.11, p=0.30). Further statistical analyses show that the memory performance change over the 2-year is not significantly related to the MPFC Aβ burden (main effect of Aβ burden: F_1,60_=2.33, p=0.13; main effect of age: F_1,60_=1.27, p=0.26, main effect of sex: F_1,60_=0.03, p=0.87, main effect of education: F_1,60_=0.41, p=0.53). When considering baseline performance to the MST across the entire sample (N=100; i.e. assessed at the same time as the sleep measures), we found no significant link between the coupling of the spindles onto both SW types and the performance on the task *main effect of spindle-SW coupling*: F_1,96_ ≥ 0.2, p ≤ 0.61; main effect of age: F_1,96_ ≤ 0.36, p ≥ 0.55, main effect of sex: F_1,96_ ≤ 0.54, p ≥ 0.46, main effect of education: F_1,96_ ≤ 0.47, p ≥0.50 (Figure 2b-c).

We modified the method section as follows (PAGE 20):

“Our primary analysis tested in a single GLMM whether the phase of coupling, as dependent variable, would differ between SW types and would be associated with Aβ burden (independent variables), controlling for sex, age, TST and SW duration. The phase of spindle-SW coupling was set as the phase of the onset of the spindle on the SW converted to its cosine value, to deal with the circularity of the phase variable and perform linear statistics (analysis using the phase in degrees yielded the same outcome). As we tested our hypothesis in a single test, it did not require correction for multiple testing and significance was set at p <.05. Our main exploratory analysis was to assess whether spindle-SW coupling to a specific SW type was associated with longitudinal change in the performance to the memory test, controlling for sex, age, education and TST. The remaining analyses were aimed a better characterising the primary analysis or the main exploratory analyses.”

The discussion draws the attention of the reader to the fact that the exploratory analyses (including cognitive decline) require replication (PAGE 14):

“Also, effect sizes of the significant associations we find were relatively small, in line with the complex and multifactorial nature of AD. In addition, the longitudinal aspect of our study is relatively short-termed and only concerned the performance to a sensitive mnesic task while it did not include sleep EEG and the PET assessments. Further studies should replicate our findings and evaluate the predictive value of such parameters on longer longitudinal protocols, and the evolution of the sleep EEG and the PET parameters as well as their generalisability over other precociously impacted cognitive abilities.”

Reviewer #1 (Recommendations for the authors):Though the data support the authors' claim, it still remains unclear whether the uncoupling of spindles and slow switcher-SWs is the earliest marker since the authors did not analyze the 0.6-1 Hz frequency band used in reference 6. To show the uncoupling is an earlier marker than 0.6-1 Hz SWA, either correspondence of slow switchers to slow oscillation (0.6-1 Hz) or direct relationships among 0.6-1 Hz power, Aβ burden, and cognitive decline should be tested.

This comment refers to essential comment (2). We did report in our initial submission that the ratio between slower (.5-1Hz) and faster δ (1-4Hz), as done in Mander et al., was not associated with early Aβ burden (Page 7, line 134, of original text). Please refer to essential comment (2) for the changes made in the text.

We stress that we are not focussing on the same frequencies as Mander et al., which are based on the arbitrary definition of a Slow Oscillation vs. a δ oscillation, but rather about the frequency of the down-to-up state transition which we distinguish between slower and faster transition frequencies based on an objective feature. We are focussing on a functional distinction between the two types of slow waves. We slightly edited the discussion to emphasise this point (PAGE 12):

“Sleep SWs are classically divided into slow oscillations and δ waves based on whether their overall frequency is lays between ~.5 and 1Hz or between 1 and 4 Hz, respectively. Although meaningful, the definition of these frequency bands was arguably arbitrary. SWs were recently divided into two categories based on an objective functional feature consisting in the frequency of their transition from the down- to the up-state, which reflects the relative synchronisation of the depolarisation of the neurons when generating a SW.”

Reviewer #2 (Recommendations for the authors):– How is 'SW type' entered into the model? Does this report the % of SWs of a single type for each participant?

SW type was entered as a class variable in the GLMM (0, 1 for slow and fast switch respectively). The variable did not depend on the % of SW of each type.

– The methods for spindle detection are important for assessing and understanding the paper and it would be helpful to describe at least briefly in the Methods, even if it repeats prior work.

The text has been modified accordingly (PAGE 17):

“Sleep spindles were also automatically detected over the same N2 and N3 epochs with a previously published method 49–51. Sleep spindles were also automatically detected over the same N2 and N3 epochs with a previously published method. The EEG signal was bandpass filtered between 10 and 16Hz with a linear phase finite impulse response filter (-3dB at 10 and 16Hz). The envelope amplitude of the Hilbert transform of this band-limited signal was smoothed and a threshold was set at the 75th percentile. All events of duration between 0.5 and 3 s were then selected as a spindle.”

– Table 1 should define what is reported in +/- vs. brackets.

A footnote has been added to the table (page 16, line 325):

“Average values ± standard deviation [range: min – max values].”

– I think the periods in the numbers for the first section of Results are intended to be commas, to indicate thousands?

The reviewer is correct and the text has been modified accordingly (page 5).

– For the difference between fast and slow SWs, it would be helpful to either direct statistically test this, or else to make more cautious statements about the two SW types and acknowledge this limitation.

Please refer to main issue 2 and 5.

Reviewer #3 (Recommendations for the authors):Suggestions:– Clarify exactly what are the primary analyses of this study, explain why these are the primary outcomes, and adjust for multiple comparisons accordingly. The primary analyses and primary conclusions, should, ideally, be aligned.

The text has been modified to take this comment into account. Please refer to Essential comment (3) for the full reply.

– Explain in greater detail how the primary measure of spindle-slow wave coupling (phase angle) was decided upon, rather than, for instance, percent coincidence, or dispersion of phase angle as a measure of the "tightness" of coupling.

Please see response to comment 3 above.

– Temper conclusions of causality given the observational nature of the work, and discuss in greater depth the likelihood that alterations in phase relationship may be markers of early AD-related brain changes, not picked up on by amyloid PET (e.g. amyloid oligomers, or non-amyloid processes).

The text has been modified accordingly. Please see response to comment 4 for full details.

– To what extent do relations with cognitive decline differ between those with high and low baseline amyloid burden?

We computed the suggested analyses for completeness (see below) but feel it may not be appropriate. Aβ burden is low in all our participants and it may not be clinically relevant to further stratify our sample. In addition, adding a regressor to our models (median split creating a lower and higher group) will create statistical penalty as it would reduce the degrees of freedom of our model. Likewise, testing another model could increase the multiple comparison issue.

For a complete response to the reviewer, we computed a GLMM using cognitive decline of extreme quartile only and Ab burden and quartile type as regressor. The analysis yielded no significant difference between extreme quartiles (p=.55), when controlling for age (p=.63), age (p=.87) and education (p=.2)